# PARAMETERS AS EXPERTS: ADAPTING VISION MODELS WITH DYNAMIC PARAMETER ROUTING

## ABSTRACT

Adapting vision models using parameter-efficient fine-tuning (PEFT) remains challenging, as it aims to achieve performance comparable to full fine-tuning using a minimal number of trainable parameters. When applied to complex dense prediction tasks, existing methods exhibit limitations, including input-agnostic modeling and redundant cross-layer representations. To address these limitations, we propose AdaRoute, a new adapter-style method featuring a simple mixture-of-experts (MoE) architecture. Specifically, we introduce shared expert centers, where each expert is a trainable parameter matrix. During a feedforward pass, each AdaRoute module in the network dynamically generates weight matrices tailored for the current module via a simple dynamic parameter routing mechanism, which selectively aggregates parameter matrices in the corresponding expert center. Dynamic weight matrices in AdaRoute modules facilitate low-rank adaptation in an input-dependent manner, thus generating more customized and powerful feature representations. Moreover, since AdaRoute modules across multiple network layers share the same expert center, they improve feature diversity by promoting implicit cross-layer feature interaction. Extensive experiments on diverse vision tasks demonstrate the superiority of AdaRoute. For instance, in the object detection and instance segmentation task on COCO2017 with ConvNeXt-L, AdaRoute significantly exceeds full fine-tuning by 1.4%/1.6% in $AP^b$/$AP^m$ using less than 5% of the trainable parameters. In the more challenging panoptic segmentation task, when Swin-B and ConvNeXt-B are used as the backbone, AdaRoute remarkably improves over AdaptFormer by 1.7% and 2.0% in PQ, respectively, while using a comparable number of trainable parameters.

## 1 INTRODUCTION

Parameter-efficient Fine-tuning (PEFT) aims to update or embed only a small number of parameters into a pre-trained model while performing comparably to full fine-tuning (Han et al., 2024). This approach has been widely adopted in both natural language processing (NLP) and computer vision. For instance, prompt-based tuning in NLP tasks (Liu et al., 2023) has inspired many PEFT methods in vision. A representative work is VPT (Jia et al., 2022), which inserts a set of learnable tokens into the input sequence of Vision Transformers (ViTs) (Dosovitskiy et al., 2021; Liu et al., 2021), achieving task adaptation with minimal additional parameters. Although prompt-based tuning methods demonstrate promising performance on classification tasks, there still exists a considerable performance gap between these methods and full fine-tuning on more complex vision tasks, such as dense predictions. On the other hand, adapter-based PEFT methods (Houlsby et al., 2019) have also attracted considerable attention. A well-known example is LoRA (Hu et al., 2022), which learns low-rank adapters to achieve very efficient fine-tuning of large language models (LLMs). In the same spirit, AdaptFormer (Chen et al., 2022) introduces a lightweight MLP module to adapt ViTs, representing an early attempt to utilize adapters in visual recognition. LoRand (Yin et al., 2023) further explores the potential of adapter-based tuning for more complex dense prediction tasks. Recently, Mona (Yin et al., 2025) integrates multi-scale depthwise convolutions into the adapter module to enhance its spatial modeling capacity for dense predictions. Although existing adapter-based methods have achieved promising results in a variety of vision tasks, two important challenges remain unresolved:

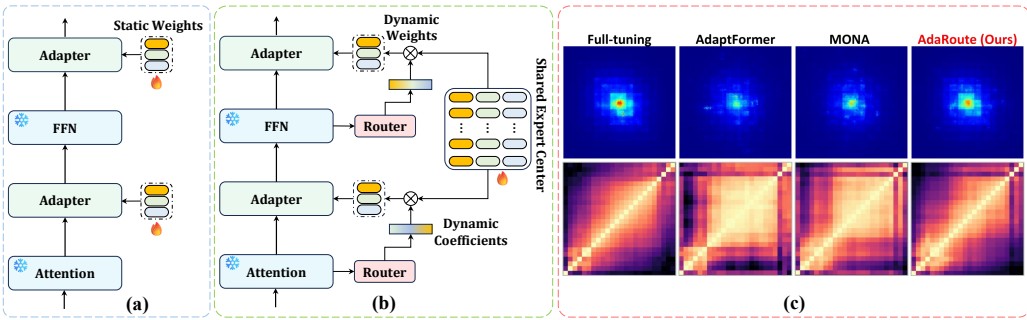

Figure 1: **(a)** Classical adapter-based PEFT methods (e.g., Mona). **(b)** Our proposed AdaRoute. Normalization layers and residual connections are omitted for simplicity. **(c)** The first and second rows show ERF and CKA visualizations for various fine-tuned models, respectively. Swin-L pre-trained on ImageNet-21K is used as the backbone network, which is fine-tuned on the COCO2017 validation set using various fine-tuning methods and the Mask R-CNN framework.

**Representation Deficiency**. As shown in Figure 1 (a), each adapter is responsible for task-specific model adaptations using input-agnostic low-rank modeling. For complex tasks such as dense predictions, it is inherently challenging to learn task-specific transformations that work universally well for all possible inputs. The fact that such adapters cannot support dynamic input-dependent adaptations to account for input variations limits the feature representation capacity of the resulting model. To empirically verify this, we use the effective receptive field (ERF) (Luo et al., 2016) to visualize a model's representation capacity. Specifically, we fine-tune the Swin-L model (Liu et al., 2021) pre-trained on ImageNet-21K (Deng et al., 2009) on the COCO2017 dataset (Lin et al., 2014) using the Mask R-CNN framework (He et al., 2017). As shown in the first row of Figure 1 (c), previous representative PEFT methods, including AdaptFormer and Mona , exhibit smaller ERFs compared to full fine-tuning. This phenomenon indicates that these methods weaken the model's ability to capture complex spatial dependencies required for dense prediction tasks.

**Feature Redundancy**. Adapters embedded in different network layers and their parameters are isolated from each other, and such a lack of cross-layer interaction may lead to redundant feature representations. To visually illustrate this limitation, we perform a centered kernel alignment (CKA) analysis (Kornblith et al., 2019) for different methods. As shown in the second row of Figure 1 (c), the patterns learned by different layers in both AdaptFormer and Mona exhibit a higher similarity compared to those under full fine-tuning, which means that different layers capture redundant information.

To address these limitations, we propose a new adapter-based PEFT method dubbed AdaRoute. As illustrated in Figure 1 (b), AdaRoute is built upon a simple mixture-of-experts (MoE) architecture (Cai et al., 2025). Specifically, we construct a large shared expert center comprising a collection of trainable parameter matrices, each having the same size as the corresponding weight matrix in a standard adapter. Each AdaRoute module in the network dynamically generates weight matrices tailored for the current module via a dynamic parameter routing mechanism, which selectively aggregates parameter matrices in this shared expert center. This routing mechanism is analogous to the gating mechanism in MoE that selects appropriate experts for a given input, and the trainable parameter matrices are treated as experts in this work. Although our design is simple, it offers two advantages that are absent in previous works:

First, dynamic weight matrices in AdaRoute modules facilitate low-rank adaptation in an input-dependent manner, thus generating more customized and powerful feature representations. As evidenced in Figure 1 (c), the ERF of our model is larger than those of other PEFT methods and comparable to that of full fine-tuning. Such a large ERF enables our model to capture long-range dependencies more easily, which is crucial in dense predictions (Xie et al., 2021).

Second, since the same expert center is shared among AdaRoute modules in multiple network layers, an implicit cross-layer feature interaction can be developed, thus reducing feature redundancy (Huang et al., 2017; Lou et al., 2025). As evidenced in Figure 1 (c), due to AdaRoute, the feature diversity of our fine-tuned model is better than those of other PEFT methods and very close

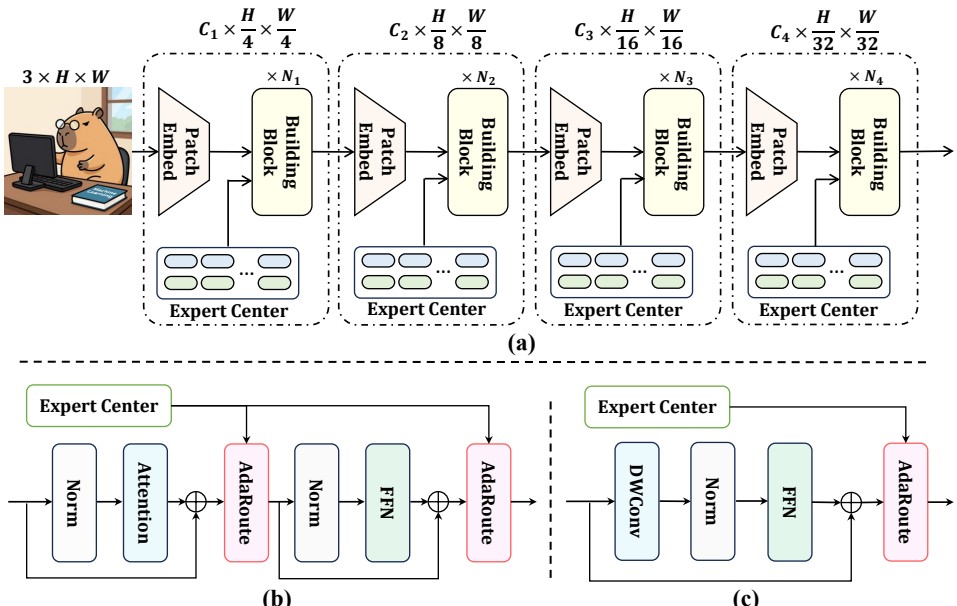

Figure 2: An overview of our proposed AdaRoute.

to that of full fine-tuning. This means that our method can extract more representative features from complex scenes for dense predictions.

We have evaluated our method on a wide range of visual recognition tasks, including object detection and instance segmentation, semantic segmentation, panoptic segmentation, and image classification. Extensive experiments in Section 4 demonstrate that our method has achieved superior performance compared to previous PEFT methods. For instance, in the semantic segmentation task on ADE20K with Swin-L, AdaRoute surpasses full fine-tuning by 0.8% in mIoU while requiring less than 4% of trainable parameters. In the object detection and instance segmentation task on COCO2017 with ConvNeXt-L, our method achieves improvements of 0.6%/0.4% in $AP^b$/$AP^m$ over Mona, using a comparable number of parameters, and outperforms full fine-tuning by 1.4%/1.6% in $AP^b$/$AP^m$. Furthermore, we employ PEFT methods for panoptic segmentation, a more challenging task that unifies semantic segmentation, object detection, and instance segmentation, and has been under-explored in prior work. Specifically, when integrated with Swin-B and ConvNeXt-B, our method improves over AdaptFormer by 1.7% and 2.0% in PQ, respectively.

## 2 RELATED WORK

**Efficient Transfer Learning in Language Models**. With the development of LLMs, PEFT techniques have been largely pioneered within the NLP community. For instance, BitFit (Ben-Zaken et al., 2022) only updates bias terms in a backbone network and parameters outside the backbone. Prompt-based tuning methods (Lester et al., 2021; Li & Liang, 2021; Liu et al., 2022a) aim to achieve task adaptation by prepending a small number of learnable tokens to the input sequence, while keeping the pre-trained weights frozen. Adapter-based methods (Houlsby et al., 2019; Pfeiffer et al., 2020; Hu et al., 2022; Liu et al., 2024) embed small trainable modules within the layers of a frozen pre-trained network. A highly influential approach is LoRA (Hu et al., 2022), which approximates weight updates via low-rank matrices. Subsequently, MoELoRA (Luo et al., 2024) employs contrastive learning to encourage distinct feature learning among experts, mitigating random routing issues. HydraLoRA (Tian et al., 2024) decomposes a projection matrix into multiple mini-rank matrices and uses a router to combine their outputs. DoRA (Liu et al., 2024) decomposes pre-trained weights into magnitude and direction to enhance both learning capacity and stability. HiRA (Huang et al., 2025) devises a Hadamard product-based LoRA to facilitate high-rank adaptation.

**PEFT for Visual Recognition**. The aforementioned PEFT methods in NLP have served as a primary source of inspiration for PEFT methods in computer vision (Jie & Deng, 2022; Luo et al., 2023; Jie & Deng, 2023). For example, VPT (Jia et al., 2022) has successfully adapted prompt-based tuning

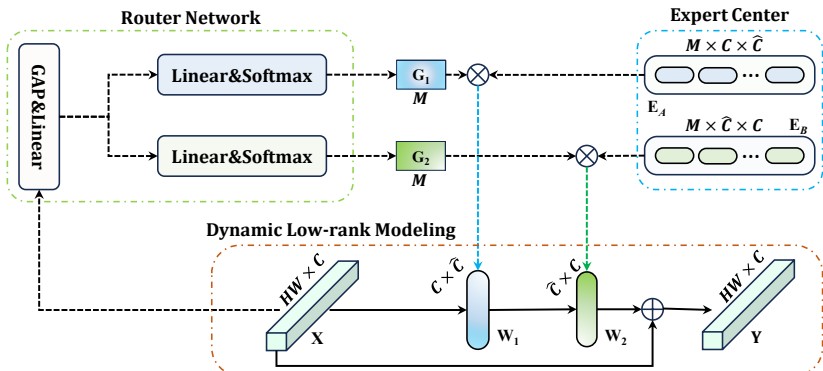

Figure 3: The workflow of dynamic parameter routing in AdaRoute.

by prepending learnable tokens to the input sequence of ViTs. DA-VPT (Ren et al., 2025) further improves visual prompt learning through semantic metric construction between prompts and image features. TCPA (Liu et al., 2025) assigns coordinated prompts to different tokens to facilitate attention interactions. On the other hand, adapter-based methods have also been extensively explored. AdaptFormer (Chen et al., 2022) attaches a parallel lightweight MLP module to the original channel mixer in ViTs. KAdaptation (He et al., 2023b) decomposes and updates adapter weights through the Kronecker product. SPT (He et al., 2023a) adaptively selects the most sensitive weights for a given task. LoRand (Yin et al., 2023) sparsely combines low-rank weights in adapters for dense predictions. Recently, Mona (Yin et al., 2025) introduces multi-scale spatial modeling capability into adapters by integrating multi-kernel convolutions.

Unlike the aforementioned methods, this paper is inspired by the MoE mechanism, but redesigns its core components: *we treat parameter matrices themselves as experts*. Specifically, we construct a large shared expert center that can be dynamically queried by layer-specific routers embedded in network layers. This enables the construction of adapters based on a rich parameter space to facilitate cross-layer interactions, resulting in superior performance over previous PEFT methods on diverse vision tasks.

## 3 METHOD

### 3.1 OVERVIEW

As illustrated in Figure 2 (a), we take a typical four-stage hierarchical vision architecture as an example, where each stage consists of a patch embedding layer followed by a few network building blocks. Within each stage, we set up a large shared expert center containing a collection of trainable parameter matrices, while AdaRoute is integrated into every building block. For a Swin-like transformer network (Liu et al., 2021), an AdaRoute module is attached after every token mixer as well as every channel mixer, as shown in Figure 2 (b). In a ConvNeXt-style network (Liu et al., 2022c), since each block encapsulates the token mixer and channel mixer into a single residual module, an AdaRoute module is attached after every complete ConvNeXt block, as shown in Figure 2 (c). During a forward pass, each AdaRoute module selectively combines parameter matrices from the shared expert center in the corresponding stage via a dynamic parameter routing mechanism, analogous to the simple expert selection strategy in MoE. This process generates dynamic weight matrices, enabling input-dependent low-rank transformation of input features.

### 3.2 ADAROUTE

**Shared Expert Center**. Each shared expert center contains a collection of trainable parameter matrices. For channel-wise transformations, the parameter matrices form pairs,

$$\left\{ \mathbf{E}_A \in \mathbb{R}^{M \times C \times \hat{C}}, \quad \mathbf{E}_B \in \mathbb{R}^{M \times \hat{C} \times C} \right\},$$

where $M$ denotes the capacity of the expert center. Both $M$ and $\hat{C}$ are hyperparameters that control the number of trainable parameters, and their configurations are discussed in Section 4.5. Since

multi-scale spatial mixing is essential for adapting models in dense prediction tasks (Yin et al., 2025), for generating dynamic weight matrices for spatial transformations, we introduce another set of parameter matrices for implementing multi-kernel depthwise convolutions,

$$\left\{ \mathbf{S}_A \in \mathbb{R}^{M \times \hat{C} \times K_1^2}, \quad \mathbf{S}_B \in \mathbb{R}^{M \times \hat{C} \times K_2^2}, \quad \mathbf{S}_C \in \mathbb{R}^{M \times \hat{C} \times K_3^2} \right\},$$

where $K_i^2$ represents one of the three kernel sizes. We discuss the combined settings of multiple kernel sizes in Section 4.5.

**Dynamic Parameter Routing**. For simplicity, let us consider the case involving channel projection only. As illustrated in Figure 3, given an input feature $\mathbf{X} \in \mathbb{R}^{HW \times C}$ (where $C$ and $HW$ denote the channel and spatial dimensions, respectively), a lightweight router network generates dynamic coefficients for the parameter matrices in the expert center. Specifically, the input feature first undergoes global average pooling (GAP), followed by a linear layer that produces a hidden representation with a significantly reduced channel dimension (e.g., 24 channels in our implementation) to minimize computational overhead. This hidden feature is then passed through two parallel linear layers with softmax activation, yielding two dynamic gating vectors $\{\mathbf{G_1}, \mathbf{G_2}\} \in \mathbb{R}^M$. These gating vectors are used to dynamically aggregate the parameter matrices in the expert center: $\mathbf{G_1}$ is multiplied with $\mathbf{E}_A$ to produce the dynamic down-projection weight matrix $\mathbf{W}_1 \in \mathbb{R}^{C \times \hat{C}}$, and similarly, $\mathbf{G_2}$ is multiplied with $\mathbf{E}_B$ to form the dynamic up-projection weight matrix $\mathbf{W}_2 \in \mathbb{R}^{\hat{C} \times C}$. Although more advanced MoE routing mechanisms exist, this simple yet efficient design aligns well with the efficiency consideration of PEFT. The dynamically composed weight matrices $\mathbf{W}_1$ and $\mathbf{W}_2$ are then used to transform the input feature $\mathbf{X}$ in an input-dependent and channel-wise manner. The final output $\mathbf{Y}$ is obtained by adding a residual connection (He et al., 2016) to this dynamically transformed input feature.

**Dynamic Multi-scale Spatial Mixing**. Inspired by Mona (Yin et al., 2025), we equip AdaRoute with multi-kernel depthwise convolutions to enhance the spatial mixing of latent features produced by the dynamic down-projection weight matrix $\mathbf{W}_1$. Nonetheless, the convolution kernels in AdaRoute are dynamically generated from the shared expert center. Specifically, the router network produces three dynamic gating vectors $\{\mathbf{G}_A, \mathbf{G}_B, \mathbf{G}_C\} \in \mathbb{R}^M$, which are multiplied with the corresponding parameter matrices $\{\mathbf{S}_A, \mathbf{S}_B, \mathbf{S}_C\}$ to produce three dynamic convolution kernels. Then, these kernels are applied to the latent features via depthwise convolutions. Although the kernel generation process is conceptually similar to classical dynamic convolutions (He et al., 2019), a key difference is that our method produces dynamic depthwise convolution kernels (D2Convs) rather than standard convolution kernels.

On the other hand, our design employs a sequentially stacked multi-scale convolution structure that progressively expands the receptive field, as depicted in Figure 4. Specifically, the input feature is sequentially fed into three D2Conv layers with increased kernel sizes. Each convolution stage is equipped with residual connections to facilitate gradient flow and preserve original information. The multi-scale outputs are then aggregated via a lightweight Spatially-varying Aggregation (SA) module, inspired by spatial attention mechanisms (Li et al., 2025). The SA module uses a $1 \times 1$ convolution

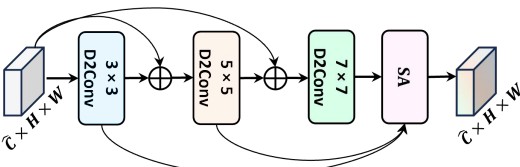

Figure 4: A schematic diagram of dynamic multi-scale spatial mixing.

followed by a softmax function to generate spatial attention maps corresponding to each scale. Each attention map is multiplied element-wise with one of the three convolutional feature maps, dynamically recalibrating these features in a spatially adaptive manner. This simple design further enhances the dynamic capacity of AdaRoute with negligible parameter overhead, leading to more powerful feature representations.

# 4 EXPERIMENTS

In this section, we present comprehensive experimental evaluations on various vision tasks, including semantic segmentation, object detection, instance segmentation, panoptic segmentation, and image classification. Recent work (Mai et al., 2025) has provided a detailed analysis of PEFT meth-

Table 1: Comparison of semantic segmentation performance on the ADE20K dataset. # P denotes the number of tunable parameters. Since all methods employ the same segmentation head, only the number of trainable parameters in the backbone network are reported.

| Method | Swin-B | | Swin-L | |
|---|---|---|---|---|
| | # P (M) | mIoU | # P (M) | mIoU |
| Full-tuning | 86.8 | 50.2 | 195.0 | 51.2 |
| VPT | 0.1 | 48.0 | 0.2 | 49.9 |
| LoRA | 5.4 | 49.4 | 8.1 | 51.1 |
| AdaptFormer | 5.4 | 50.0 | 8.1 | 51.3 |
| LoRand | 5.9 | 49.9 | 8.2 | 51.4 |
| Mona | 5.2 | 49.8 | 7.5 | 51.6 |
| **AdaRoute** | 5.2 | **50.3** | 7.3 | **52.0** |

| Method | ConvNeXt-B | | ConvNeXt-L | |
|---|---|---|---|---|
| | # P (M) | mIoU | # P (M) | mIoU |
| Full-tuning | 87.6 | 51.4 | 196.2 | 52.4 |
| AdaptFormer | 6.4 | 50.3 | 9.6 | 50.9 |
| LoRand | 6.7 | 49.6 | 9.3 | 51.2 |
| Mona | 6.5 | 50.7 | 9.1 | 51.5 |
| **AdaRoute** | 6.5 | **51.1** | 9.2 | **52.0** |

ods for the image classification task, but this work focuses primarily on dense prediction tasks, as efficient dense prediction is more challenging to achieve and has more real-world applications. All experiments are conducted on 4 NVIDIA H800 GPUs.

**Pre-trained Models and Baselines**. For dense prediction tasks, we employ two representative hierarchical vision backbone networks, including Swin and ConvNeXt. In particular, we use the base and large versions pre-trained on ImageNet-21K for both backbone architectures. Regarding image classification, we employ ViT-B/16 (Dosovitskiy et al., 2021) pre-trained with MAE (He et al., 2022), adhering to the setting in Chen et al. (2022). This setup allows us to comprehensively validate the generalization ability and robustness of our method. Meanwhile, we also measure the performance of other representative baseline methods, including VPT (Jia et al., 2022), LoRA (Hu et al., 2022), AdaptFormer (Chen et al., 2022), LoRand (Yin et al., 2023), and Mona (Yin et al., 2025), on the same tasks using the same pre-trained backbones. For a fair comparison, we adjust the latent dimension of adapter-based methods to ensure that they have a comparable number of trainable parameters. We retain the original configuration of VPT since introducing additional prompt tokens would lead to substantial computational overhead due to self-attention operations. Since LoRA and VPT are specialized transformer-based PEFT methods, they are not applied to ConvNeXt.

## 4.1 SEMANTIC SEGMENTATION

**Setup**. Semantic segmentation experiments are conducted on the ADE20K dataset (Zhou et al., 2017) using the UperNet framework (Xiao et al., 2018). We adhere to the experimental setting in Swin (Liu et al., 2021), where all models are trained for 160K iterations using the AdamW optimizer (Loshchilov & Hutter, 2019) with a "poly" learning rate schedule (Chen et al., 2017) and a batch size of 16.

**Results**. Table 1 shows that our method achieves leading performance in semantic segmentation compared to the baselines. Specifically, using Swin-B as the backbone, AdaRoute achieves the highest mIoU of 50.3%, which is slightly better than the performance of full fine-tuning, while saving approximately 95% of the trainable parameters. Meanwhile, when using Swin-L as the backbone, AdaRoute achieves a notable performance improvement of 0.8% over full fine-tuning while only using less than 4% of the parameters. Furthermore, compared to Mona, AdaRoute achieves performance increases of 0.5% and 0.4% using Swin-B and Swin-L, respectively. AdaRoute also significantly surpasses LoRand by 1.5% and 0.8% in mIoU when using ConvNeXt-B and ConvNeXt-L, respectively, and performs on par with full fine-tuning while using less than 8% of the parameters.

Table 2: Comparison of object detection and instance segmentation performance on the COCO2017 dataset.

| Method | Swin-B | | | | | | Swin-L | | | | | |
| | # P (M) | $AP^b$ | $AP^b_{50}$ | $AP^b_{75}$ | $AP^m$ | $AP^m_{50}$ | $AP^m_{75}$ | # P (M) | $AP^b$ | $AP^b_{50}$ | $AP^b_{75}$ | $AP^m$ | $AP^m_{50}$ | $AP^m_{75}$ |
|---|---|---|---|---|---|---|---|---|---|---|---|---|---|---|
| Full-tuning | 86.8 | 47.5 | 69.8 | 52.3 | 42.8 | 66.6 | 46.0 | 195.0 | 48.6 | 70.9 | 53.7 | 43.8 | 68.1 | 47.3 |
| VPT | 0.1 | 40.6 | 65.7 | 44.0 | 38.8 | 62.7 | 41.3 | 0.2 | 42.6 | 67.8 | 46.1 | 40.5 | 64.9 | 43.1 |
| LoRA | 5.4 | 40.1 | 65.1 | 43.2 | 38.5 | 62.1 | 41.0 | 8.1 | 42.3 | 67.6 | 46.2 | 40.4 | 64.6 | 43.5 |
| AdaptFormer | 5.4 | 43.9 | 67.8 | 48.0 | 40.8 | 65.0 | 43.8 | 8.1 | 46.3 | 70.2 | 50.9 | 42.8 | 67.0 | 46.0 |
| LoRand | 5.9 | 42.8 | 67.0 | 46.5 | 40.2 | 64.0 | 43.0 | 8.2 | 44.9 | 69.2 | 49.2 | 41.8 | 66.1 | 45.0 |
| Mona | 5.2 | 46.6 | 69.4 | 50.9 | 42.4 | 66.2 | 45.6 | 7.5 | 48.1 | **71.3** | 52.8 | 43.9 | 68.2 | 47.6 |
| **AdaRoute** | 5.2 | **47.3** | **70.0** | **51.4** | **42.7** | **66.7** | **46.2** | 7.3 | **48.6** | 71.1 | **53.4** | **44.0** | **68.4** | **47.6** |

| Method | ConvNeXt-B | | | | | | ConvNeXt-L | | | | | |
| | # P (M) | $AP^b$ | $AP^b_{50}$ | $AP^b_{75}$ | $AP^m$ | $AP^m_{50}$ | $AP^m_{75}$ | # P (M) | $AP^b$ | $AP^b_{50}$ | $AP^b_{75}$ | $AP^m$ | $AP^m_{50}$ | $AP^m_{75}$ |
|---|---|---|---|---|---|---|---|---|---|---|---|---|---|---|
| Full-tuning | 87.6 | 47.8 | 69.7 | 52.4 | 43.0 | 66.9 | 46.3 | 196.2 | 48.1 | 69.7 | 53.1 | 43.2 | 66.8 | 46.7 |
| AdaptFormer | 6.4 | 44.8 | 68.4 | 49.0 | 41.6 | 65.4 | 44.7 | 9.6 | 45.8 | 69.7 | 50.1 | 42.5 | 66.4 | 45.8 |
| LoRand | 6.7 | 43.9 | 67.6 | 47.7 | 41.0 | 64.6 | 44.4 | 9.3 | 45.1 | 68.8 | 49.4 | 42.0 | 65.8 | 45.2 |
| Mona | 6.5 | 47.5 | 70.0 | 52.2 | 43.2 | 67.0 | 46.7 | 9.1 | 48.9 | 71.4 | 53.7 | 44.4 | 68.4 | 48.1 |
| **AdaRoute** | 6.5 | **48.0** | **70.2** | **52.7** | **43.5** | **67.5** | **46.9** | 9.2 | **49.5** | **71.9** | **54.4** | **44.8** | **69.2** | **48.4** |

## 4.2 Object Detection and Instance Segmentation

**Setup**. Performance in object detection and instance segmentation is evaluated on the COCO2017 dataset (Lin et al., 2014) using the Mask R-CNN framework (He et al., 2017), following the same experimental setting in Swin (Liu et al., 2021). Specifically, all models are trained for 12 epochs using the AdamW optimizer (Loshchilov & Hutter, 2019) with a batch size of 16.

**Results**. Unlike semantic segmentation, object detection and instance segmentation are more challenging, since models are required not only to predict bounding boxes but also to output instance-level segmentation masks, which test the spatial modeling ability of PEFT methods more directly. As shown in Table 2, our method delivers impressive performance compared to other PEFT methods. For instance, using Swin-B, AdaRoute improves over Mona by 0.7%/0.3% in $AP^b$/$AP^m$. When using Swin-L, AdaRoute achieves remarkable performance improvements of 3.7%/2.8% in $AP^b$/$AP^m$ over LoRand while performing on par with full fine-tuning. When using ConvNeXt, AdaRoute exceeds all other PEFT methods and full fine-tuning. Specifically, AdaRoute improves over full fine-tuning by 0.2%/0.5% in $AP^b$/$AP^m$ using ConvNeXt-B. When using ConvNeXt-L, AdaRoute significantly outperforms full fine-tuning by 1.4%/1.6% in $AP^b$/$AP^m$.

## 4.3 Panoptic Segmentation

**Setup**. We further perform evaluations on panoptic segmentation (Kirillov et al., 2019). This task unifies semantic segmentation, instance segmentation, and object detection, providing a more comprehensive evaluation of a model's dense prediction capability. Experiments are conducted on the COCO2017 dataset using the Panoptic FPN framework (Kirillov et al., 2019). All models are trained for 12 epochs using the AdamW optimizer (Loshchilov & Hutter, 2019) with a batch size of 16. Performance metrics include panoptic quality (PQ), segmentation quality (SQ), and recognition quality (RQ).

**Results**. Table 3 provides the quantitative results of various models. Compared to other PEFT methods, our method delivers notable performance improvements. For instance, AdaRoute significantly surpasses AdaptFormer by 1.7% and 1.1% in PQ using Swin-B and Swin-L, respectively. Similarly, using ConvNeXt variants, AdaRoute improves over Mona by 0.6% and 0.9% in PQ. However, we notice that our method still has a moderate performance gap with full fine-tuning in this task. The reason might be that the extreme parameter efficiency of PEFT methods (less than 8% of the trainable parameters of full fine-tuning) imposes fundamental limits on their ability to capture the highly complex and diverse features required for panoptic understanding. The concurrent demands of instance-level segmentation for things and semantic segmentation for stuff in panoptic segmentation necessitate a higher representation capability, which might not be fully attainable through PEFT alone. Nevertheless, our method still exceeds all baselines.

Table 3: Comparison of panoptic segmentation performance on the COCO2017 dataset.

| Method | Swin-B | | | | Swin-L | | | |
| | # P (M) | PQ | SQ | RQ | # P (M) | PQ | SQ | RQ |
|---|---|---|---|---|---|---|---|---|
| Full-tuning | 86.8 | 50.3 | 81.3 | 60.6 | 195.0 | 51.4 | 81.5 | 61.9 |
| VPT | 0.1 | 45.1 | 78.6 | 55.4 | 0.2 | 46.6 | 79.1 | 57.0 |
| LoRA | 5.4 | 45.3 | 78.5 | 55.7 | 8.1 | 46.6 | 79.4 | 57.1 |
| AdaptFormer | 5.4 | 47.1 | 79.4 | 57.4 | 8.1 | 48.8 | 79.9 | 59.2 |
| LoRand | 5.9 | 46.4 | 79.3 | 56.7 | 8.2 | 47.7 | 80.3 | 58.0 |
| Mona | 5.2 | 48.1 | 79.9 | 58.3 | 7.5 | 49.7 | 80.7 | 60.2 |
| **AdaRoute** | 5.2 | **48.8** | **80.8** | **59.0** | 7.3 | **50.2** | **81.3** | **60.5** |

| Method | ConvNeXt-B | | | | ConvNeXt-L | | | |
| | # P (M) | PQ | SQ | RQ | # P (M) | PQ | SQ | RQ |
|---|---|---|---|---|---|---|---|---|
| Full-tuning | 87.6 | 50.2 | 81.2 | 60.5 | 196.2 | 51.0 | 80.9 | 61.4 |
| AdaptFormer | 6.4 | 46.6 | 78.8 | 56.7 | 9.6 | 47.2 | 79.6 | 57.3 |
| LoRand | 6.7 | 45.9 | 78.6 | 56.0 | 9.3 | 46.8 | 79.4 | 56.9 |
| Mona | 6.5 | 48.3 | 80.1 | 58.5 | 9.1 | 49.5 | 80.9 | 59.7 |
| **AdaRoute** | 6.5 | **48.9** | **81.0** | **59.1** | 9.2 | **50.4** | **81.2** | **60.7** |

Table 4: Comparison of image classification performance obtained using ViT-B/16.

| Method | # P (M) | CIFAR-100 | SVHN | Food-101 | Avg. |
|---|---|---|---|---|---|
| Full-tuning | 86.0 | 85.9 | 97.7 | 90.1 | 91.2 |
| VPT | 0.1 | 82.4 | 94.0 | 83.0 | 86.5 |
| LoRA | 3.9 | 86.2 | 97.1 | 87.8 | 90.4 |
| AdaptFormer | 4.3 | 86.2 | 97.0 | 87.9 | 90.4 |
| LoRand | 3.9 | 86.1 | 96.9 | 87.9 | 90.3 |
| Mona | 3.9 | 87.0 | 97.3 | 89.6 | 91.3 |
| **AdaRoute** | 3.9 | **87.7** | **97.7** | **89.7** | **91.7** |

## 4.4 IMAGE CLASSIFICATION

**Setup**. Following Chen et al. (2022), we conduct image classification experiments on three widely adopted datasets: CIFAR-100 (Krizhevsky et al., 2009), SVHN (Netzer et al., 2011), and Food-101 (Bossard et al., 2014). The experimental setup follows the standard configuration as detailed in Chen et al. (2022). Specifically, all models are trained for 100 epochs with the cosine learning rate schedule (Loshchilov & Hutter, 2017) with 20 warm-up epochs and a total batch size of 1024.

**Results**. Table 4 reports the top-1 accuracy of different methods on each dataset and their average performance on all three datasets, and it can be found that our method delivers the best performance among all methods considered. AdaRoute achieves an improvement of 0.5% in average accuracy over full fine-tuning. In comparison to the best-performing baseline (Mona), AdaRoute improves by 0.7%, 0.4%, and 0.1% in top-1 accuracy. The modest gains are consistent with prior observations (Mai et al., 2025) that different PEFT methods often yield comparable performance in low-shot image classification tasks, likely because the decision boundaries in such settings are relatively simple to learn. In contrast, dense prediction tasks require more diverse knowledge and can better demonstrate the capability of a model in handling complex feature representations.

## 4.5 ABLATION STUDIES

**Setup**. We conduct comprehensive ablation studies on object detection and instance segmentation, utilizing Swin-B pre-trained on ImageNet-21K as the backbone network. The remaining experimental settings follow the configuration described in Section 4.2. Due to page limits, more experimental evaluations are presented in the Appendix.

**Trade-off between Latent Dimension and Expert Center Capacity**. We investigate the effect of two key hyperparameters: the expert center capacity $M$ and the latent dimension $\hat{C}$. To enable easy integration of AdaRoute into various network architectures, we set $M$ according to the number of layers $L$ in each stage of the network. For example, in stage 3 of Swin-B, which has 18 layers,

Table 5: Impact of latent dimension and expert center capacity.

| Trade-off | # P (M) | $AP^b$ | $AP^m$ |
|---|---|---|---|
| $[M = 4L, \hat{C} = 40]$ | 5.5 | 46.1 | 42.0 |
| $[M = 2L, \hat{C} = 72]$ | 5.4 | 46.9 | 42.4 |
| $[M = L, \hat{C} = 128]$ | 5.2 | **47.3** | **42.7** |
| $[M = \frac{L}{2}, \hat{C} = 192]$ | 5.4 | 47.2 | 42.6 |

Table 6: Impact of expert center scope. Trainable parameter counts of all models are omitted due to negligible differences.

| # Layers/group | $AP^b$ | $AP^m$ |
|---|---|---|
| 18 | **47.3** | **42.7** |
| 9 | 47.1 | 42.5 |
| 6 | 46.5 | 42.0 |
| 3 | 46.3 | 41.9 |

setting $M = 2L$ yields $M = 36$. Table 5 reveals that performance is governed by a balance between latent dimension and expert diversity. Optimal results occur at $[M = L, \hat{C} = 128]$, indicating that an overly small latent dimension or expert center capacity results in performance degradation.

**Effect of Shared Expert Center**. We evaluate the effectiveness of the shared expert center by varying its scope within the network. Specifically, in stage 3 of Swin-B (18 layers), we divide the expert center into smaller ones, each shared within a smaller group of consecutive layers. We experiment with groups of 3, 6, and 9 consecutive layers. Note that stages 1, 2, and 4 are not divided due to their small number of layers (2 layers only). As shown in Table 6, performance degrades as the groups become smaller. Notably, when expert centers are shared within small groups (e.g., 3 or 6 layers), the performance becomes comparable to Mona, indicating that the benefit of AdaRoute primarily comes from the proposed large shared expert center design.

**Effect of Kernel Sizes in Multi-scale Spatial Mixing**. We investigate the effect of kernel sizes of depthwise convolutions in Section 3.2. As shown in Table 7, using a single kernel size results in suboptimal performance, as it fails to capture object contexts at different scales. In contrast, using multiple kernel sizes yields better results. Specifically, the combination of kernel sizes $[3, 5, 7]$ achieves the best performance. However, further increasing the kernel sizes does not lead to better performance, which we attribute to the difficulty of learning with large kernel sizes under low-rank conditions. Furthermore, incorporating the spatially varying aggregation (SA) module leads to modest performance improvements with a negligible increase in trainable parameters.

Table 7: Effect of kernel sizes in multi-scale spatial mixing.

| Kernel Size | # P (M) | $AP^b$ | $AP^m$ |
|---|---|---|---|
| 3 | 5.1 | 46.3 | 42.2 |
| 5 | 5.2 | 46.4 | 42.2 |
| 7 | 5.4 | 46.6 | 42.5 |
| $[3, 5, 7]$ | 5.2 | 47.1 | 42.6 |
| $[5, 7, 9]$ | 5.5 | 46.9 | 42.6 |
| $[3, 5, 7] + $ SA | 5.2 | **47.3** | **42.7** |
| $[5, 7, 9] + $ SA | 5.5 | 47.2 | **42.7** |

## 5 LIMITATIONS

Although our AdaRoute achieves better performance than other PEFT methods in diverse vision tasks, and sometimes even surpasses full fine-tuning using an extremely reduced number of trainable parameters, it still has limitations. Like representative vision adapters such as AdaptFormer and Mona, our AdaRoute cannot be integrated into the network during inference. In addition, the introduction of multi-scale depthwise convolutions results in a minor increase in training latency. Furthermore, due to resource constraints, we do not evaluate our approach on larger-scale models, such as SwinV2-G (Liu et al., 2022b). In our future work, we aim to further improve efficiency, reduce the number of trainable parameters, and achieve even stronger performance, while also expanding evaluations to larger models.

## 6 CONCLUSION

In this paper, we propose AdaRoute, a new PEFT method for adapting vision models. Inspired by expert routing in MoE, our method constructs a large shared expert center where trainable parameter matrices serve as experts. A lightweight dynamic routing mechanism aggregates these experts to generate input-dependent weights for the network, thus improving cross-layer feature interaction and feature representation quality. Extensive experimental results on multiple challenging vision tasks demonstrate that AdaRoute achieves superior performance compared to existing PEFT methods.

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

# A APPENDIX

## A.1 MORE ABLATION STUDIES

Building on the training settings outlined in Section 4.5, we present additional ablation experiments to systematically dissect the contribution of each component in our proposed method.

**Impact of Different Initialization Methods of Expert Center**. We adopt `trunc_normal` as the default initialization strategy for the expert centers. To further investigate the impact of different initialization methods, we also evaluate `kaiming_normal` and `kaiming_uniform`. As shown in Table 8, results demonstrate that our method is not sensitive to the choice of parameter initialization, that is, different strategies lead to only negligible differences in final performance, which confirms the training robustness of our approach.

Table 8: Impact of different parameter initialization methods.

| Init. Method | $AP^b$ | $AP^m$ |
|---|---|---|
| `trunc_normal` | **47.3** | **42.7** |
| `kaiming_normal` | 47.1 | 42.6 |
| `kaiming_uniform` | 47.2 | 42.7 |

Table 9: Effect of the layout of multi-kernel dynamic convolutions.

| Method | $AP^b$ | $AP^m$ |
|---|---|---|
| Parallel | 47.1 | 42.6 |
| Sequential (w/o Res.) | 46.6 | 42.3 |
| Sequential (w Res.) | **47.3** | **42.7** |

Table 10: Effect of activation function in dynamic routing.

| Activation | $AP^b$ | $AP^m$ |
|---|---|---|
| Sigmoid | 46.2 | 41.9 |
| Softmax | **47.3** | **42.7** |

**Other Design Choices**. We further examine several design choices in AdaRoute. First, we evaluate the layout of multi-scale dynamic convolutions introduced in Section 3.2. As shown in Table 9, our sequential layout of multi-kernel convolutions with residual connections yields better performance compared to both a residual-free sequential layout and a parallel layout without introducing extra trainable parameters. We also test the activation functions used in the router. Table 10 indicates that the use of sigmoid activation leads to performance degradation, as it does not adequately model the

Table 11: Comparison of efficiency at the input resolution of $1280 \times 800$ with a batch size of 4.

| Method | # P (M) | Thr. (imgs/s) | $AP^b$ | $AP^m$ |
|---|---|---|---|---|
| Full-tuning | 195.0 | 4.63 | 48.6 | 43.8 |
| LoRA | 8.1 | 4.07 | 42.3 | 40.4 |
| AdaptFormer | 8.1 | 5.15 | 46.3 | 42.8 |
| LoRand | 8.2 | 5.11 | 44.9 | 41.8 |
| Mona | 7.5 | 3.94 | 48.1 | 43.9 |
| **AdaRoute (w/o Conv)** | 7.0 | 4.96 | 47.7 | 43.5 |
| **AdaRoute** | 7.3 | 3.96 | **48.6** | **44.0** |

competition among experts in the shared center. In contrast, softmax activation provides a better probability distribution over experts.

## A.2 COMPUTATIONAL EFFICIENCY ANALYSIS

To evaluate computational efficiency, we measure training throughput (Thr.) using Swin-L as the backbone. We focus solely on the efficiency of the backbone network since PEFT modules are only integrated into the backbone. Specifically, the input size is set to $1280 \times 800$ with a batch size of 4, and measurements are averaged over more than 100 iterations using a single NVIDIA L40S GPU based on the COCO2017 dataset. As shown in Table 11, our method achieves an excellent trade-off among the number of trainable parameters, throughput, and performance. In particular, our method matches the speed of the best-performing baseline while delivering clearly better performance. Although our approach is approximately 15% slower than full fine-tuning during training, this is due to the use of multi-kernel convolutions in the adapter design, following Mona. To analyze this overhead, we construct a simplified variant, termed AdaRoute (w/o Conv), by removing all multi-kernel convolutions. Results show that this variant achieves comparable throughput to AdaptFormer, while significantly improving $AP^b$ and $AP^m$ by 1.4% and 0.7%, respectively.

On the other hand, full fine-tuning requires storing a separate version of the network parameters for each downstream task, while each version is nearly as large as the original pre-trained model. This leads to significant storage overhead when multiple tasks are carried out. In contrast, our AdaRoute requires less than 4% of the trainable parameters to achieve comparable performance, reducing the storage cost by more than 95%.

## THE USE OF LARGE LANGUAGE MODELS

In preparing this paper, the authors utilized DeepSeek-R1 (Guo et al., 2025) and Llama-3.1 (Grattafiori et al., 2024) to improve the readability and grammatical accuracy of selected texts, subsequent to their initial drafting.

## REPRODUCIBILITY STATEMENT

We have provided implementation details in this paper. Although the code is not included with this paper submission, we are committed to making the source code publicly available once the paper receives a final decision.

## ETHICS STATEMENT

This work proposes a generic transfer learning method for efficient vision recognition. All experiments are conducted on publicly available datasets and utilize open-source pre-trained models, thereby ensuring the absence of private or sensitive data. Furthermore, our research is designed to be domain-agnostic and does not pose any ethical concerns, as it is not specifically tailored towards potentially harmful application domains, including surveillance or misinformation dissemination.

