# OpenReview forum: "Parameters as Experts: Adapting Vision Models with Dynamic Parameter Routing"
_ICLR.cc/2026/Conference — ICLR 2026 Conference Withdrawn Submission_

### Official Review · Reviewer_nt45 · 2025-10-18

**Soundness:** 3
**Presentation:** 3
**Contribution:** 2
**Rating:** 4
**Confidence:** 4

**Summary:**

This paper proposes AdaRoute, a parameter-efficient fine-tuning framework that treats parameter matrices as experts within a shared expert center. Each network layer uses a lightweight router to dynamically combine these experts, generating input-dependent low-rank adaptations. This design improves representation flexibility and cross-layer feature diversity while remaining efficient. Experiments on multiple vision benchmarks show that AdaRoute outperforms existing PEFT methods and even surpasses full fine-tuning in dense prediction tasks with less than 5% of trainable parameters.

**Strengths:**

1. The proposed router-expert structure is simple yet effective. It avoids the heavy computational cost of standard MoE while introducing input-conditioned flexibility.
2. By sharing a global expert center among AdaRoute modules, the method enables implicit cross-layer communication through jointly updated expert matrices. This design effectively reduces feature redundancy and encourages representation diversity.
3. The paper conducts comprehensive experiments on various vision tasks, including semantic segmentation, object detection, instance segmentation, panoptic segmentation, and image classification, demonstrating consistently superior performance over existing PEFT methods and even surpassing full fine-tuning on several dense prediction benchmarks.

**Weaknesses:**

1. The paper compares AdaRoute with only a limited set of baseline methods. It lacks evaluation against several state-of-the-art PEFT approaches widely used for vision model adaptation, such as MLAE [1], SPT [2], DA-VPT [3], RepAdapter[4]. Including these baselines would provide a more convincing and comprehensive comparison.
2. Conceptually, the proposed method can be viewed as an adapter-based architecture augmented with a Mixture-of-Experts (MoE) gating mechanism. Although the authors introduce a shared global expert center, the idea of treating parameters as experts is not entirely novel — for instance, MLAE [1] regards each rank-1 submatrix as an independent expert. Furthermore, the adoption of the Dynamic Multi-scale Spatial Mixing module, inspired by Mona, enhances performance on visual downstream tasks but does not constitute a fundamentally new contribution.
3. Since the paper does not provide any theoretical justification, the validity of the proposed approach relies entirely on experimental results. However, the experimental settings for both AdaRoute and baseline methods are insufficiently described, making it difficult to ensure reproducibility and fairness in comparison.

------

[1] MLAE: Masked LoRA Experts for Visual Parameter-Efficient Fine-Tuning. https://arxiv.org/abs/2405.18897

[2] Sensitivity-Aware Visual Parameter-Efficient Fine-Tuning. https://arxiv.org/abs/2303.08566

[3]  DA-VPT: Semantic-Guided Visual Prompt Tuning for Vision Transformers. https://arxiv.org/abs/2505.23694

[4] Towards Efficient Visual Adaption via Structural Re-parameterization. https://arxiv.org/abs/2302.08106

**Questions:**

1. While the paper introduces Dynamic Multi-scale Spatial Mixing inspired by Mona to improve performance on visual tasks, the proposed framework itself is not limited to vision-specific settings. Extending AdaRoute to large language models (e.g., LLaMA3-8B or Qwen-2.5-7B) could substantially broaden its impact and demonstrate the generality of the approach.
2. Given that AdaRoute is built upon an adapter-based design, could the authors explore integrating the proposed routing and expert-sharing mechanism into LoRA-style low-rank adaptation?

---

### Official Review · Reviewer_rk3W · 2025-10-28

**Soundness:** 3
**Presentation:** 3
**Contribution:** 3
**Rating:** 6
**Confidence:** 5

**Summary:**

The paper proposes a new Parameter-Efficient Eine-Tuning (PEFT) method, AdaRoute, which treats trainable parameter matrices as experts and builds a shared expert center. By employing a lightweight routing network to dynamically aggregate parameters within this shared center, the method achieves input-dependent weight generation, enhancing cross-layer feature interaction and feature representation quality.

**Strengths:**

1. The paper introduces a mixture-of-experts mechanism into the field of PEFT. By employing a routing network to aggregate shared expert matrices, it achieves input-dependent weight generation, enabling better adaptation to downstream tasks. Meanwhile, the shared expert center design implicitly enhances cross-layer feature interaction.
2. The AdaRoute module is lightweight and highly generalizable. It can dynamically generate parameters, effectively reducing redundant feature learning during fine-tuning and thereby lowering training costs.
3. The paper is well-structured and logically coherent, with comprehensive experimental design and sufficiently validated results.

**Weaknesses:**

1. The paper lacks an in-depth theoretical explanation and mathematical analysis of the proposed dynamic parameter routing mechanism. It does not clearly elaborate on the principles, optimization stability, or convergence properties underlying the method, making its theoretical foundation relatively weak.
2.  Although the introduction of multi-scale convolutions enhances spatial modeling capability, it also introduces additional computational overhead, leading to training delays. Furthermore, the paper does not provide a detailed evaluation of inference efficiency or deployment cost.

**Questions:**

The authors are encouraged to provide a more detailed theoretical explanation or mathematical analysis of the dynamic parameter routing mechanism to clarify its underlying principles and advantages over traditional methods. In addition, it is suggested to include visualization or experimental analysis illustrating the expert activation patterns under different inputs, which would further enhance the understanding and interpretability of the model’s behavior.

---

### Official Review · Reviewer_8ZCa · 2025-10-30

**Soundness:** 2
**Presentation:** 2
**Contribution:** 2
**Rating:** 2
**Confidence:** 3

**Summary:**

This paper proposes AdaRoute, which generates matrices in low-rank adapters through shared expert parameter matrices. The authors claim that this approach can enhance the model’s effective receptive field and reduce feature redundancy. Experimental results on several dense prediction tasks demonstrate the superiority of AdaRoute compared with other models.

**Strengths:**

- The AdaRoute method is simple yet effective.
- The fine-tuning of vision models on large-scale dense prediction tasks is an important research topic.

**Weaknesses:**

My main concern lies in the methodological overlap between this work and LoRand, as well as Mona, and in the lack of a strong motivation. The idea of using MoE to generate low-rank matrices has already been introduced in LoRand, while Mona enhances adapter performance in dense prediction through multi-scale convolutions. The proposed method in this paper appears to be a combination of these two approaches.

Although the authors point out the limitations of prior work in terms of effective receptive field and feature redundancy, the paper does not provide further experimental analysis to:
1. demonstrate why these two issues are critical for dense prediction, or whether their impact on model performance is indeed significant;
2. explain, from a methodological perspective, how AdaRoute is designed to mitigate these issues; and
3. verify that the performance gains of AdaRoute actually stem from addressing these problems.

Given the current content, the paper does not establish a sufficiently strong logical connection from motivation to method to results.

**Questions:**

- The authors should provide comparisons of memory overhead and computational cost with the baseline under different parameter scales.
- What are the architectural configurations of the baselines used for comparison in the paper?
- Line 228: “produces a hidden representation with a significantly reduced channel dimension” — does this refer to a reduction from C to M? The description of the `Dynamic Parameter Routing` section is somewhat vague; it is recommended that the authors include equations to illustrate the process more clearly.

---

### Official Review · Reviewer_1aKe · 2025-11-02

**Soundness:** 3
**Presentation:** 3
**Contribution:** 3
**Rating:** 4
**Confidence:** 4

**Summary:**

The paper proposes AdaRoute, a PEFT method that treats parameter matrices as experts and uses a lightweight router to compose input-dependent low-rank adapters from a shared expert center across layers. In addition to channel projections, AdaRoute generates dynamic depthwise multi-kernel convolutions to improve spatial mixing. The authors claim AdaRoute reduces representation deficiency (small ERFs) and feature redundancy (high cross-layer CKA similarity), outperforming prior adapters (AdaptFormer, LoRand, Mona) and sometimes even full fine-tuning on dense prediction tasks. Experiments on COCO2017 with ConvNeXt-L, AdaRoute reportedly exceeds performance than full fine-tuning and on ADE20K with Swin-L, AdaRoute improves mIoU by +0.8 over full fine-tuning; and on panoptic segmentation, it improves PQ over AdaptFormer by +1.7 (Swin-B) and +2.0 (ConvNeXt-B), with comparable trainable parameters.

**Strengths:**

1. **Clear problem framing with concrete evidence.** The paper motivates representation deficiency (smaller ERFs) and feature redundancy (higher cross-layer CKA) for prior adapters and illustrates the proposed fix via shared expert centers and dynamic routing (Fig. 1–2).

2. **Simple, implementable mechanism.** The expert-center + router design is straightforward; the method section specifies shapes and routing steps, and extends naturally to multi-kernel depthwise convolutions (Fig. 3–4).

3. **Strong dense-prediction results.** On ADE20K, COCO detection/segmentation, and panoptic segmentation, AdaRoute consistently matches or beats prior PEFT baselines and in some cases surpasses full fine-tuning with <~5–8% trainable parameters.

4. **Useful ablations.** The paper explores expert capacity vs. latent dimension, scope of sharing across layers, and kernel-size choices, yielding actionable guidance.

5. **Honest limitations and efficiency table.** The authors openly note training latency and inference constraints, and quantify throughput vs. baselines on Swin-L at 1280×800.

**Weaknesses:**

# Major Concerns
1. **Routing stability & expert semantics under-analyzed.**
While the router composes per-input weights from a shared center, the paper gives limited analysis of expert specialization, collapse/over-use, or routing entropy (beyond softmax vs. sigmoid). It is unclear whether distinct experts truly encode complementary functions or if routing degenerates. Consider reporting expert usage histograms, KL/entropy of gates, and CKA across experts, not only layers.

2. **Compute/memory cost of dynamic convolutions.**
AdaRoute’s best variant includes sequential multi-kernel depthwise layers with SA; efficiency analysis shows ~15% slower training than full fine-tuning (Swin-L), which is surprising for a PEFT goal. A deeper breakdown (routing MLP cost, expert aggregation, D2Conv FLOPs) and memory profile would clarify scalability on higher resolutions.

3. **Fairness nuances in baseline coverage.**
The paper excludes some transformer-specific baselines on ConvNeXt (e.g., VPT/LoRA) due to modality mismatch but then compares AdaRoute against full fine-tuning on ConvNeXt where it wins. One may question how much gain comes from architecture-specific advantages (depthwise D2Conv) vs. the routing idea itself. Please expand on why transformer-only PEFT baselines are omitted on ConvNeXt and include a ConvNeXt-compatible LoRA variant (e.g., conv-LoRA or structural re-param adapters) for apples-to-apples.

4. **Causal link between ERF/CKA and task metrics is assumed, not proven.**
Fig. 1 suggests AdaRoute’s ERF and cross-layer CKA are “closer to full tuning,” but the paper does not quantify how these correlate with AP/mIoU gains across settings. A correlation study across seeds/tasks ($\Delta$ERF/$\Delta$CKA vs. $\Delta$mIoU/AP) would bolster the representational narrative.

5. **Generalization beyond detection/segmentation is limited.**
Classification gains are modest (avg +0.5 over full-tuning on ViT-B/16), consistent with prior observations that PEFT methods cluster on low-shot classification; but this also tempers the generality claim. Adding robustness, OOD segmentation, or few-shot dense benchmarks would strengthen impact.

## Minor Concerns
1. **Terminology consistency.** Clarify “cannot be integrated into the network during inference”. Do you mean the backbone at inference uses the adapters (it should), but you cannot merge them into base weights? Wording is confusing.

2. **Ablation scope.** Nice coverage on M/Ĉ, scope, and kernels; please also ablate router width, gate temperature, and expert center per-stage vs. global sharing.

3. **Reproducibility.** Code is not provided at review time; include a checklist and promise artifacts upon decision (noted), but config files and seed scripts would help.

4. **Table**. Add “higher is better” to captions; report std for Tables 1–4; include trainable-parameter counts in Table 2–3 for quick comparison.

5. **Routing visualization.** A small UMAP of routed features or per-expert activation maps on COCO images would make the mechanism tangible.

**Questions:**

1. **Expert usage & collapse.** Do a few experts dominate? Please provide expert-selection entropy, per-expert frequency, and diversity metrics across datasets and layers.

2. **Routing robustness.** How sensitive are results to router hidden size, gate temperature, or adding noise/regularizers (e.g., entropy or load-balancing)?

3. **Compute breakdown.** What fraction of FLOPs/latency comes from (i) router, (ii) expert aggregation, (iii) D2Convs, (iv) SA? Any memory spikes due to storing expert centers?

4. **ConvNeXt fairness.** Can you include a conv-LoRA or re-param adapter baseline on ConvNeXt to ensure architectural fairness?

5. **ERF/CKA–metric connection.** Do improvements in ERF/CKA predict AP/mIoU across seeds? A correlation plot would substantiate the representation claim.

6. **Inference statement.** Please clarify “cannot be integrated during inference.” Do you simply keep AdaRoute active at inference (standard for adapters), or is there a constraint that prevents its usage?

7. **Global vs. per-stage centers.** You share centers per stage; did you test a single global center (with stage-aware conditioning) or cross-stage reuse? What is the impact on redundancy?

8. Any evidence that routing helps when class boundaries are complex (e.g., long-tail or OOD classification), beyond CIFAR/SVHN/Food-101?

---

### Note · Authors · 2025-11-14

I have read and agree with the venue's withdrawal policy on behalf of myself and my co-authors.